# Potential for New Therapeutic Approaches by Targeting Lactate and pH Mediated Epigenetic Dysregulation in Major Mental Diseases

**DOI:** 10.3390/biomedicines12020457

**Published:** 2024-02-18

**Authors:** Shabnam Nohesara, Hamid Mostafavi Abdolmaleky, Sam Thiagalingam

**Affiliations:** 1Department of Medicine (Biomedical Genetics), Boston University Chobanian and Avedisian School of Medicine, Boston, MA 02118, USA; snohesar@bu.edu; 2Nutrition/Metabolism Laboratory, Beth Israel Deaconess Medical Center, Harvard Medical School, Boston, MA 02215, USA; 3Department of Pathology & Laboratory Medicine, Boston University Chobanian and Avedisian School of Medicine, Boston, MA 02118, USA

**Keywords:** neuropsychiatric diseases, pH, lactate, lactylation, DNA methylation, histone acetylation, microbiome

## Abstract

Multiple lines of evidence have shown that lactate-mediated pH alterations in the brains of patients with neuropsychiatric diseases such as schizophrenia (SCZ), Alzheimer’s disease (AD) and autism may be attributed to mitochondrial dysfunction and changes in energy metabolism. While neuronal activity is associated with reduction in brain pH, astrocytes are responsible for rebalancing the pH to maintain the equilibrium. As lactate level is the main determinant of brain pH, neuronal activities are impacted by pH changes due to the binding of protons (H^+^) to various types of proteins, altering their structure and function in the neuronal and non-neuronal cells of the brain. Lactate and pH could affect diverse types of epigenetic modifications, including histone lactylation, which is linked to histone acetylation and DNA methylation. In this review, we discuss the importance of pH homeostasis in normal brain function, the role of lactate as an essential epigenetic regulatory molecule and its contributions to brain pH abnormalities in neuropsychiatric diseases, and shed light on lactate-based and pH-modulating therapies in neuropsychiatric diseases by targeting epigenetic modifications. In conclusion, we attempt to highlight the potentials and challenges of translating lactate-pH-modulating therapies to clinics for the treatment of neuropsychiatric diseases.

## 1. Introduction

The onset and the time course of pathophysiological processes in the brains of patients with neuropsychiatric diseases can be influenced by a variety of factors. One of the most important factors is cerebral pH, because of the high sensitivity of enzyme activity and protein folding to pH changes [1]. Several investigations have demonstrated that abnormalities in pH regulation are linked to brain pathophysiological changes and serious complications in other bodily organs [2,3]. Sodium bicarbonate or acetate are considered to be good candidates for the prevention or treatment of conditions associated with reduced pH [4,5]. A low extracellular pH in patients can alter the concentrations of Ca^2+^ ions, protein activities, and cellular metabolism. These changes may ultimately result in the loss of cellular function and/or cell death [6]. Due to the high sensitivity of various central nervous system (CNS) proteins, such as those involved in synaptic transmission, ion channels, and neurotransmitter receptors, even minimal changes in brain pH can significantly impact various signaling pathways, cellular membrane excitability, action potential generation, and synaptic activities. These effects are crucial in the pathophysiology of neuropsychiatric diseases [7]. As an example, the voltage-gated Ca^2+^ channels exhibit high sensitivity to low pH, which in turn reduces the Ca^2+^ concentration in the presynaptic elements and inhibits neurotransmitter secretion [3]. The detrimental effects of extracellular acidification can also be linked to mitochondrial depolarization, the generation of free radicals in brain synaptosomes, and hence oxidative stress in neuronal presynaptic endings [8,9]. Moreover, P2Y receptors (which respond to extracellular nucleotides, such as ATP and ADP)-mediated Ca^2+^ signaling and microglia migration are inhibited under conditions of extracellular acidosis [10].

It has been found that abnormalities in pH regulation are also associated with epigenetic aberrations linked to pathological conditions. For instance, the expression of Na^+^/H^+^ exchangers (NHEs) (like NHE-3) can be regulated by DNA methylation alterations in their genes [11]. Thus, abnormalities in or the loss of NHE gene functions through epigenetic aberrations can result in alterations of the intracellular pH in the brain, potentially contributing to the initiation and progression of neuropsychiatric diseases. Some therapeutic agents are capable of diminishing the severity of the disease by affecting epigenetic alterations of genes involved in pH regulation. For instance, estradiol treatment reduces alpha-synuclein accumulation effects in *C. elegans* by increasing NHE-2 levels. This increase is achieved through the induction of lysine methylation (H3K4me3) and promoter-specific histone H3 acetylation (H3K9) [12]. In this review, our goal is to first discuss the importance and mechanisms of maintaining pH homeostasis in normal brain functions. Next, we systematically review the relationships among lactate as an essential epigenetic regulatory molecule, pH abnormalities, and neuropsychiatric diseases. Thirdly, we discuss the therapeutic potentials of pH-modulating interventions affecting epigenetic changes in neuropsychiatric diseases. Finally, we address practical challenges and their solutions related to translating pH-modulating therapies into clinical practice for neuropsychiatric diseases.

## 2. Brain pH Homeostasis: Normal Function and Pathology

Neuronal activity and metabolism are associated with the successive production and release of protons into the extracellular space. This accumulation of acidic substances leads to a decrease in extracellular pH, further influencing neuronal functions. Due to the high sensitivity of all the biochemical and electrogenic machinery of synaptic transmission to pH changes, a precise buffering capacity via the elimination of excessive protons (H^+^) is crucial for efficient communication within neuronal circuits for the normalization of brain functions [7,13,14,15]. One of the most important mechanisms for controlling tissue pH is buffering with CO_2_/HCO_3_^−^ [16]. A dynamic equilibrium has been observed between CO_2_/H_2_O and H^+^/HCO_3_^−^ in this system. The activity of enzymes belonging to the carbonic anhydrase family is essential to maintaining this dynamic equilibrium [17]. Brain tissue is protected from acidification by bicarbonate buffering and carbonic anhydrase activity via converting H^+^ and HCO_3_^−^ to H_2_O and CO_2_. It has been found that epigenetic mechanisms play a key role in the transcriptional control of genes relevant to the carbonic anhydrase family. For example, histone deacetylase 4 (HDAC4) is involved in inhibiting the transcriptional activity of the carbonic anhydrase IX promoter, which contributes to the reversible hydration of carbon dioxide (H_2_O + CO_2_ ⇔ H^+^ + HCO_3_^−^) [18].

In the brain tissue astrocytes, a class of cells originated from the ectoderm are capable of preserving the brain milieu from acidification via normalizing local extracellular pH homeostasis through the neuronal activity-dependent production of bicarbonate [19]. High levels of electrogenic sodium bicarbonate cotransporter 1 (NBCe1, SLC4A4) are expressed by astrocytes, which can transport large amounts of HCO_3_^−^ across the astroglial membrane [20,21]. Additionally, the expression of HCO_3_^−^ transporter NBCe2 in the choroid plexus (a key structure maintaining the blood–cerebrospinal fluid (CSF) barrier functions) plays a crucial role in normalizing CSF pH during high CO_2_ production [22].

Approximately one third of all astrocytes produce bicarbonate to facilitate the buffering of neuronal activity-dependent extracellular H^+^ accumulation, which in turn helps maintain normal neuronal activity. This involves a sequential chain of events, including the activity-dependent release of ATP, triggering bicarbonate production through the activation of metabotropic P2Y1 receptors, phospholipase C recruitment, Ca^2+^ release from the internal depot sites, and the outward transportation of HCO_3_^−^ via NBCe1, which in turn helps maintain the normal neuronal activity [23]. The NHE1 or solute carrier family 9 member 1 (SLC9A1) is another electroneutral active transporter and a key player in H^+^ extrusion from the astrocytes in exchange for Na^+^ [24,25,26].

Some NHEs have also been found to be master regulators of pH in cellular organelles. For instance, as microglia are the main immune cells of the brain that play an important role in the clearance and degradation of Alzheimer amyloid fibrils (fAβ), an increase in the acidification of lysosomes is key to the enhancement of microglia capabilities as regards eliminating Alzheimer amyloid fibrils. Under normal conditions, the microglial lysosomes are less acidic (average pH of ~6 versus pH of ~5 in J774 macrophages), and they abolish the activity of lysosomal enzymes in these cells. Elevating the acidification of lysosome by interleukin-6 treatment or macrophage colony-stimulating factor (MCSF) facilitates the degradation of fAβ by microglia [27]. Notably, it has also been reported that the downregulation of NHE6 gives rise to excessive endosomal acidification in ApoE4-expressing astrocytes, which in turn results in a reduction in the clearance of amyloid beta Aβ peptide. On the contrary, HDAC inhibitors can increase NHE6 expression and Aβ clearance [28]. The transcriptional activity of NHEs can also be regulated by other epigenetic mechanisms. For instance, the downregulation of miR-135a under pathological conditions gives rise to NHE9 overexpression [29].

The pH of the endolysosomal lumen is also precisely regulated by a balance between the proton pump and “leak pathway” [30]. The electrogenic pumping of protons by the V-type H^+^-ATPase in conjunction with vesicular chloride transporters of the CLC (chloride channel genes) family is responsible for endosomal acidification [30,31]. In addition, proton leak mechanisms, including proton-coupled antiporters or proton conduction channels, help prevent acidification, balancing the pump and leak pathways, and restore the pH set point. Owing to the high transport rates (~1500 ions per second) of Na^+^/H^+^ exchangers, even slight disturbances in their expression or activity could lead to substantial changes in the ionic milieu within the limited boundaries of the endosomal lumen, which may further promote neuropsychiatric diseases [32,33,34].

## 3. Association between pH Abnormalities and Neuropsychiatric Diseases

Several lines of evidence support a strong association between brain pH irregularities and different neuropsychiatric diseases. For example, an increase in the brain lactate level associated with decreased brain pH has been reported in schizophrenia (SCZ) and bipolar disorder, supported by meta-analyses of >10 different studies [35,36,37]. An increased brain lactate level as the main determinant of reduced brain pH has also been reported in other diseases with psychotic symptoms, such autism and Alzheimer disease (AD), and in drug-naïve animal models of psychiatric diseases [36]. A recent meta-analysis of 281 human datasets for 11 brain diseases revealed that gene expression alterations associated with decreased pH were over-represented in brain disorders, including SCZ, bipolar disorder, autism spectrum disorders (ASD), AD, Huntington’s and Parkinson’s diseases. Furthermore, cell type-specific analyses identified that astrocytes express the most acidity-related genes, consistent with previous findings showing a lower astrocytic intracellular pH compared to neurons [38]. As the brain pH is lower than the blood pH (6.6–7.0 vs. 7.0–7.5), and neuronal activity produces H^+^ and decreases brain pH, astrocytes related bicarbonate production and its release (mediated by SLC4A4) are responsible for brain pH regulation (see Figure 1), which in turn resulting in the acidic pH of astrocytes [23].

It has been demonstrated that lactate increases transforming growth factor beta-2 (TGFB2) expression and anti-lactate drugs inhibit this effect [39]. Another study reported that increased TGFB2 expression in post-mortem brains of patients with SCZ and bipolar disorder is due to its promoter DNA hypomethylation [40]. Furthermore, in subsequent analyses of these data, where the brain pH in patients with SCZ and bipolar disorder was less than in control subjects (*p* = 0.03 and *p* = 0.01, respectively, 2-tail *t* test), there was an inverse correlation between TGFB2 expression and brain pH in general (r = −0.54, *p* = 0.002). Remarkably, as astrocytic genes were among the most upregulated genes in SCZ and bipolar disorder, and the expression of TGFB2 in induced pluripotent stem cells (iPSC)-derived astrocytes was three-fold higher than in neurons [40], the expression of over 75% of dysregulated genes in the disease state (in particular astrocytic genes) exhibited a direct correlation with TGFB2 expression. This suggests that a lactate-induced acidic brain pH related to the epigenetic upregulation of TGFB signaling may orchestrate the widespread gene expression dysregulation observed in SCZ and BD. To examine if lactate or pH affect TGFB2 expression in brain cells, we treated iPSC-derived neurons and astrocytes with lactic acid and HCL to decrease the pH of culture medium by 0.3 and 0.6 pH units. We found that these interventions firmly increased TGFB2 expression in a dose-dependent manner in astrocytes (unpublished data).

With reference to the importance of TGFB signaling in SCZ pathogenesis, a meta-analysis of 40 studies concluded that TGFβ is “a disease state marker in SCZ”. In fact, “its expression is increased in acutely relapsed SCZ patients” as well as first-episode psychosis, but it is normalized by antipsychotic treatment [41]. Regarding other mental diseases, while TGFB2 was the highest-ranked risk gene in a deep post-GWAS analysis of AD patients [42], its increased expression has been reported in post-mortem brains of patients with AD [42,43]. AD also exhibits reduced brain pH due to higher lactate levels. It is important to note that brain pH remains stable after death (0–48 h) and during freezer storage [44]. Therefore, it is unlikely that the relationship between reduced brain pH and neuropsychiatric diseases is due to post-mortem preservation conditions. In Table 1 and Table 2 we have summarized the results of other studies that largely support the link between reduced brain pH due to increased brain lactate and the pathogenesis of major mental diseases.

More studies addressing an association between elevated lactate levels and neuropsychiatric diseases are summarized in Table 2.

## 4. Association between Lactate, an Essential Epigenetic Regulatory Molecule, and Neuropsychiatric Diseases

Lactate plays crucial roles in numerous physiological processes in humans, including acting as a signaling molecule that influences the immune system, as well as serving as an energy source and a pH regulator. At basic pH, lactate is in the form of sodium salt (sodium lactate), and at low pH it is in the protonated acidic form (lactic acid) [71]. It has been shown that lactate administration is capable of creating specific exercise-induced brain alterations and enhancing memory formation by improving synaptic functions [72,73,74,75]. In fact, exercise-induced lactate contributes to switching from the pro-inflammatory to the anti-inflammatory phenotype in microglia through histone lactylation, and hence prevents cognitive dysfunctions [76]. Based on “lactate timer” theory, increased lactate levels in inflammatory-stimulated macrophages can induce histone modifications like H3 lactylation, and thereby affect the transcription of specific genes relevant to the process of tissue repair [77]. Treatment with exogenous lactate also enhances histone lactylation and elevates arginase 1 (ARG1) expression, an indicator of a reparative phenotype [78]. Therefore, lactate appears to be a master regulator of a metabolic–epigenetic link in macrophage polarization by increasing histone acetylation [79]. However, the production of excessive amounts of lactate for a prolonged period of time in neuropsychiatric and other hard-to-treat diseases may result in the acidification of the extracellular microenvironment to a pH range of 6.0 to 6.5 [80]. Acidosis is capable of promoting angiogenesis, inflammation, and immunosuppression [81,82]. Since a lower pH of the brain tissue in neuropsychiatric diseases has been attributed to mitochondrial dysfunction and lactate accumulation [83], the lactylation of histone proteins (a newly recognized epigenetic modification) has the capacity to promote chromatin accessibility in a fashion that relies on oxidative energy metabolism [78,84]. In reality, histone lactylation through modifying gene transcription plays key roles in some biological processes, including the reprogramming of somatic cells and tumorigenesis [78,85,86]. Several recent studies have also shown that lactate may play crucial roles in the pathogenesis of brain diseases by increasing histone lactylation, and hence dysregulating genes expression patterns [87,88]. For instance, it has been shown that an elevated glycolysis-derived lactate level could accelerate histone 3 lysine 9 lactylation (9H3K9la) at the promoter of solute carrier family 7 member 11 (SLC7A11) and confer microglial activation and neuroinflammation by increasing its transcription in Parkinson’s disease [89]. Therefore, the inhibition of glycolysis is considered to be a promising strategy to prevent the detrimental effects of lactate in neuropsychiatric disorders, especially SCZ. As another interesting example, Xie et al. recently found that glycolysis and lactylation were elevated in the MK801-induced SCZ model in vivo and in vitro, and treatment with 2-DG (a glycolysis inhibitor) mitigated behavioral changes in a SCZ mice model by alleviating lactate accumulation and histone H3K9 and H3K18 lactylation [90].

In addition to histone acetylation, an association between DNA methylation and disrupted mitochondrial function has been linked to increased brain lactate in patients with ASD [91]. Sun et al. demonstrated that Rett syndrome (caused by underlying mutations in MeCP2 (Methyl-CpG binding protein 2)) could alter the astrocytes’ DNA methylation landscape and corresponding transcriptional features of genes, which in turn results in abnormal energy metabolism, mitochondrial dysfunction, and the secretion of high levels of lactate [92]. MeCP2 is a key protein with binding capacity to various types of methylated DNA in the brain, and its mutations could highly influence important cellular and molecular signatures of human astrocytes during the maturation process, including changes in lactate production [93]. The ability of medullary astrocytes that sense PCO_2_/[H^+^] changes is also regulated by MeCP2. In fact, medullary astrocytes play key roles in central CO_2_/pH chemosensitivity through MeCP2, and its deficiency reduces the ability of medullary astrocytes to sense PCO_2_/[H^+^] changes [94]. It has been found that MECP2-associated mitochondrial dysfunction is also linked to elevated ergometric lactate levels and can induce autistic-like features, learning disabilities, and cognitive decline, indicating a key role for the MECP2 gene in the mitochondrial pathways [95]. Moreover, a recent study showed that human astrocytes with MECP2 mutations produce more lactate and have abnormal energy metabolism due to mitochondrial dysfunction [92]. According to the results of a single-base-pair-resolution sequencing of the DNA methylome, treatment with L-lactate or D-lactate resulted in DNA hyper- or hypomethylation of many CpG sites in neuronal cells [96], confirming that an elevated lactate level can induce widespread brain DNA methylation changes.

Lactate-derived epigenetic modifications also aggravate the microglial dysfunction and neuroinflammation involved in AD development [97]. Despite its roles in the pathogenesis of neuropsychiatric diseases (Table 2), lactate can serve as a promising candidate for pH-modulating therapies, since it is capable of promoting adult hippocampal neurogenesis [98,99]. It has been reported that the neuroprotective effects of lactate against some neuropsychiatric diseases are also associated with epigenetic mechanisms. In an interesting example, Karnib et al. found that short-term lactate use in male C57BL/6 mice before any exposure to daily stress (for ten days) could act as an antidepressant agent via increasing resilience to stress and restoring normal levels and the activity of some class I HDACs (HDAC2 and HDAC3) [100]. However, as shown in Table 2, based on multiple human and animal studies, the long-term effects of higher levels of lactate may be deleterious for neurodevelopment, potentially leading to neuropsychiatric diseases.

## 5. pH-Modulating Effects of Neuropsychiatric Drugs via Epigenetic Changes

Clinically relevant doses of antipsychotics and antidepressants in vitro give rise to neuronal intracellular pH changes, which may partially be responsible for their actions. While a moderate acidification may reduce neuronal activity, Bonnet et al. have shown that a short-term use of micro-molar levels of different antidepressants (e.g., trimipramine, amitriptyline, citalopram, mirtazapine, and doxepin) and antipsychotics (e.g., clozapine, haloperidol, and ziprasidone) could reversibly decrease intracellular pH in the neurons of hippocampal slices by up to 0.35 pH units [101]. However, in a six-week clinical study, Jensen et al. examined the neuroprotective effects of triacetyluridine (TAU), a dietary supplement, against depressive symptoms in patients with bipolar disorder, and found that TAU, by increasing brain pH, is capable of suppressing symptoms of depression [102]. In another study, Cai et al. reported an elevated plasma level of lactate in first-episode neuroleptic-naïve SCZ patients [103]. They found that as the plasma lactate level in drug-free patients was higher than in control subjects, six weeks of treatment with risperidone, an atypical antipsychotic drug, further increased its level, which was suggested to be a compensatory mechanism for energy deficiency (because of a compromised glucose metabolism) in SCZ patients. In a study by Elmorsy et al., patients were treated with classic antipsychotic drugs (chlorpromazine or haloperidol) or atypical antipsychotics (risperidone, olanzapine, or quetiapine) over a three-month period. Their results reveal that treatment with chlorpromazine and haloperidol caused a noticeable elevation in lactate levels within the first 10 days of therapy, whereas all antipsychotics markedly increased arterial blood lactate levels after 90 days compared to the baseline levels [104]. While these two studies measured blood lactate level, another study using proton magnetic resonance spectroscopy demonstrated that treatment with quetiapine for 12 weeks in patients with bipolar disorder resulted in a reduction in brain lactate levels, which was associated with improvements in their clinical symptoms [105]. It has been reported that antidepressants are also capable of suppressing symptoms of major depression by regulating the brain pH homeostasis and by reducing astrocytic dysfunction [106]. Hence, owing to their key role in brain pH homeostasis (via bicarbonate and protons transport), astroglial cells are considered key cellular targets for the treatment of neuro-psychiatric diseases [107,108]. In another interesting study, Ren et al. reported that treatment with fluoxetine could elevate astroglial intracellular pH in a dose- and time-dependent fashion by stimulating NHE1 and by the outward transportation of protons [106]. Their results show that fluoxetine treatment (at 1–10 μM concentrations that is used in clinical settings) for three and four weeks in vitro could elevate astrocytes’ intracellular pH from 7.05 to 7.34 and from 7.18 to 7.58, respectively [106]. Notably, altered interplays and distributions of organellar NHEs, which are involved in protein trafficking, along with the reduced expression of NHE8, have been found in the post-mortem dorsolateral prefrontal cortex of SCZ patients. These findings were not related to antipsychotic use [109], suggesting the potential application of fluoxetine as an adjuvant therapy in refractory SCZ patients.

In patients with bipolar disorder, an elevated gray matter lactate level and reduced intracellular pH (resulting from the lactate acidosis) have been reported in the frontal lobes and in the basal ganglia of medication-free patients [110,111,112]. Lithium is one of the most effective drugs in bipolar disorder, as it improves mitochondrial function via epigenetic changes. In line with this notion, Scola et al. found that rotenone could reduce the production of ATP and complex I activity and increase apoptotic cell death via elevating the levels of hydroxymethylcytosine (5-hmc) and 5-methylcytosine (5-mc) in rat cortical primary neurons. However, treatment with lithium could reduce cell death by hampering rotenone-induced mitochondrial complex I dysfunction through changing 5-mc and 5-hmc levels [113]. Another study showed that lithium is able to reduce mitochondrial damage and apoptosis by preventing the imbalances in histone acetyl transferase (HAT) and HDAC activities [114]. In addition, the protective effects of lithium have been linked to the intracellular alkalinization of astrocytes and the modulation of their functions. For instance, the chronic treatment with lithium at concentrations used in clinical settings elevated intracellular pH in astrocytes [115]. The acute stimulation of NHE (an H^+^/Na^+^ exchanger gene) using extracellular lithium could also lead to intracellular alkalinization in astrocytes [115]. Another study demonstrated that the chronic treatment of primary cultures of normal mouse astrocytes with three conventional anti-bipolar disorder drugs (lithium, carbamazepine and valproic acid) resulted in a gradual intracellular alkalinization via various mechanisms [116].

In regard to neurodegenerative diseases, as an increased nuclear translocation of HDAC4 in ApoE4 carrier astrocytes was associated with the downregulation of NHE6 expression, broad-spectrum HDAC inhibitors (like sodium valproate) were able to increase NHE6 expression, normalize endosomal pH, and mitigate Aβ clearance defects in ApoE4 astrocytes [28]. Other types of brain cells such as microglia, a group of immune cells resident in the brain that play key roles in neuroinflammation, are also affected by brain lactate or pH and exhibit epigenetic alterations. For example, in aged as well as AD mice models (FAD4T and APP/PS1), it has been shown that the elevated levels of lactic acid in senescent microglia of the hippocampus tissue could increase the level of pan histone lysine lactylation (Kla) and histone 3 lysine 18 lactylation (H3K18la). H3K18la in turn stimulates the nuclear factor kappa B (NF-κB) signaling pathway through binding to the promoter of p65 and NFκB1 (p50), which results in the overexpression of IL-6 and IL-8, the senescence-associated cytokines involved in brain aging and AD [117]. Therefore, more studies are needed not only to understand the contributions of lactate pH to the brain’s non-neuronal cells (e.g., microglia and various types of astrocytes) and their epigenetic regulatory mechanism, but also to uncover the potential effects of new or known therapeutics, which could affect the brain lactate pH level in neuropsychiatric diseases.

As shown in Figure 2, several drugs that are commonly used in brain or psychiatric diseases may affect the brain’s pH. For instance, topiramate and levetiracetam, by inhibiting carbonic anhydrase function, could reduce the intracellular pH of hippocampal neurons, thus exerting protective effects against epilepsy [118]. Both of these drugs (or their major metabolites) are also epigenetic modifiers [119,120]. Additionally, it has been shown that acetazolamide, another carbonic anhydrase inhibitor, used in idiopathic intracranial hypertension (and as an adjuvant drug in refractory epilepsy) is able to increase the choroid plexus intracellular pH and HCO_3_^−^ in three week-old Sprague-Dawley rats [121]. Several studies have also shown that spironolactone as another inhibitor of carbonic anhydrases (I, II and IV) exhibiting beneficial effects in neuropsychiatric diseases like SCZ, depression, and autism [122,123,124].

## 6. Lactate Producing and Utilizing Gut Microbiota and Their Roles in pH Modulation

Besides the neuropsychiatric drugs that affect lactate and pH levels, there are several types of gut microorganisms that are capable of producing lactic acid via the fermentation of carbohydrates or indigestible fibers. Common species of lactic acid bacteria (LAB) are *Lactobacillus* (e.g., *Lactobacillus acidophilus*, *Lactobacillus casei*), *Leuconostoc*, *Lactococcus*, *Pediococcus*, and *Streptococcus* (e.g., *Streptococcus thermophilus*, *Streptococcus salivarius*) [125]. LAB can be employed as probiotics to exert their beneficial health effects since they are able to produce short-chain fatty acids (SCFAs), vitamins, amines, and exopolysaccharides [126]. Based on the bacterial strains and pH, several lactate-utilizing bacteria are capable of converting lactate into SCFAs (e.g., butyrate, acetate, valerate, propionate), which inhibit HDACs and exhibit health-promoting effects [127,128]. Some SCFAs, like butyrate and acetate, are well known epigenetic modifiers and anti-inflammatory agents that contribute to the improvement of neuropsychiatric diseases such as SCZ and depression [129,130,131,132].

Lower numbers of gut lactate-utilizing bacteria can create a sequential chain of events, including lactate accumulation, the reduced production of butyrate and propionate, and drastic changes in microbiota composition [133]. Low pH (~5.5) contributes to an accelerated periodic lactate accumulation, causing harmful effects [133]. Hence, lactate-utilizing bacteria can be used as probiotics to balance pH and reduce acute inflammatory responses [134]. Their protective effects against mental diseases such as depression, AD, and autism are associated with improving the intestinal microenvironment, maintaining intestinal barrier integrity, the overexpression of brain-derived neurotrophic factor (BDNF), and the modulation of mucosal immunity and the proportion of beneficial bacteria in the intestine [135,136]. Moreover, lactate-utilizing bacteria are capable of suppressing symptoms of neuropsychiatric diseases via increasing concentrations of epigenetic modifiers such as butyrate and acetate. For example, the protective effects of p62 (SQSTM1)-engineered LAB in AD are linked to their increasing abundance in the family of *Ruminococcacea* (a major producer of butyrate), reducing neuronal oxidative stress and inflammation, regulating the ubiquitin–proteasome system and autophagy, and attenuating amyloid peptide levels [137].

A recent systematic review of more than 40 studies, including ~2500 cases and ~2400 control subjects, concluded that higher levels of lactic acid-producing bacteria in the gastrointestinal tract were associated with neuropsychiatric diseases such as SCZ, MDD and bipolar disorder [138]. Moreover, it has been shown that gut microbiota-related lactate production in mammals could result in adverse effects like acidosis and neurotoxicity [139]. Nonetheless, elevated lactate production in the brain of SCZ patients may lead to oral microbiome changes in favor of the lactate-utilizing genera of bacteria as a compensatory mechanism offering potentially helpful remedies in SCZ [140].

An elevated abundance of LAB (*Limosilactobacillus* and *Lactobacillus*) and the reduced abundance of bacteria that produce SCFAs (*Paraprevotella* and *Faecalibacterium*) in SCZ patients have also been connected to lifelong psychological stress and poor dietary habits [141]. In another study, the higher abundance of LAB, including *Bifidobacterium* and *Lactobacilli,* has been linked to the induction of chronic inflammation in individuals with SCZ [142]. In addition, AD pathology has been linked to increased levels of LAB and decreased concentrations of acetate and butyrate [143]. Altogether, since intestinal lactate can cross gut–blood and blood–brain barriers and affect brain functions [144], these findings indicate that the use of probiotics and/or prebiotics that limit lactate production or increase lactate utilization could be considered as potential remedies in the treatment of brain diseases that are already overwhelmed and exhausted by higher lactate production (Table 2).

## 7. Opportunities and Challenges in Translating pH-Modulating Therapies to the Clinic for the Treatments of Neuropsychiatric Diseases

Multiple lines of evidence indicate that lactate pH alterations and neuroinflammation in neuropsychiatric diseases may lead to gene expression patterns linked to the epigenetic alterations of brain cells, including non-neuronal cells of the brain. However, there are some conflicting findings that warrant further investigations in this field of research. For example, a recent study using iPSC-derived neurons suggests that the loss-of-function mutations of SET domain containing 1A (SETD1A), a histone methyltransferase, are associated with higher risk of SCZ and reduced levels of lactate release in the supernatant of cultured neurons, which was proposed to be due to the disruption of glycolytic functions [145]. However, one can argue that this could be due to the lack of an efficient outward lactate transport mechanism (e.g., due to SLC4A4 downregulation) that may perturb neuronal functions. In fact, as shown in Table 2, many studies have indicated a strong link between an elevated lactate level and cognitive dysfunctions in patients with SCZ and other major psychiatric diseases, including in drug-naïve patients or in animal models of psychiatric diseases. Controversial findings can be attributed to various technical limitations, such as an underpowered study design, and the restriction of the study to a specific brain region such as the hippocampus. For example, one study found an elevated pH level in the left hippocampus of patients with AD [146], whereas another study reported no significant difference in the pH of the hippocampus between subjects with AD and controls [53], hinting that more attention to potential unilateral alterations is required during data analysis. Conflicting findings may also bring up the question of whether the increased level of lactate in mental diseases contributes to the disease pathogenesis, or is a compensatory response to the disease state.

Regarding the potential effects of psychiatric drugs played out through cellular pH regulation, the exact mechanisms by which antipsychotics and antidepressants may exert neuroprotection via pH changes are unclear. While some antipsychotic drugs may decrease (rather than increase) brain pH, several investigations have shown that other neuropsychiatric drugs are capable of restoring the balance of lactate production, thus its epigenetic impacts. Other lines of evidence indicate that their neuroprotective functions may also come from inducing the alkalinization of the astroglia cytosol, which affects neuronal functions. The emergence of single-cell analysis technologies, like single cell RNA and DNA sequencing or other new methods related to single cell expression and epigenetic profiling, have paved the way to accelerate the translation of lactate pH-modulating transcriptome and epigenetic alterations to the clinic, and will provide additional insights into the pathophysiological functions of different brain cells involved in the pathogenesis of neuropsychiatric diseases in coming years. However, it is important to note that pH dynamics are primarily homeostatic in nature, involving various feedback loops. Therefore, the long-term therapeutic targeting of pH levels may also lead to adverse effects.

## 8. Conclusions

Brain pH due to alterations in the level of lactate, which may be linked to mitochondria dysfunction and aberrations in energy metabolism, have substantial impacts on cell signaling cascades, membrane excitability, synaptic activities, and cognitive functions. The lactate pH irregularities in neuropsychiatric diseases are associated with epigenetic aberrations, especially changes in DNA methylation and histone acetylation, and the newly discovered histone lactylation which are linked to altered gene expression patterns. pH-modulating therapies may be considered promising strategies for improving neuropsychiatric diseases by restoring the balance in the production of lactate or inducing the alkalinization of astroglia cytosol via transcriptional regulatory mechanisms or nutritional, microbiome and pharmacological interventions.

## Figures and Tables

**Figure 1 biomedicines-12-00457-f001:**
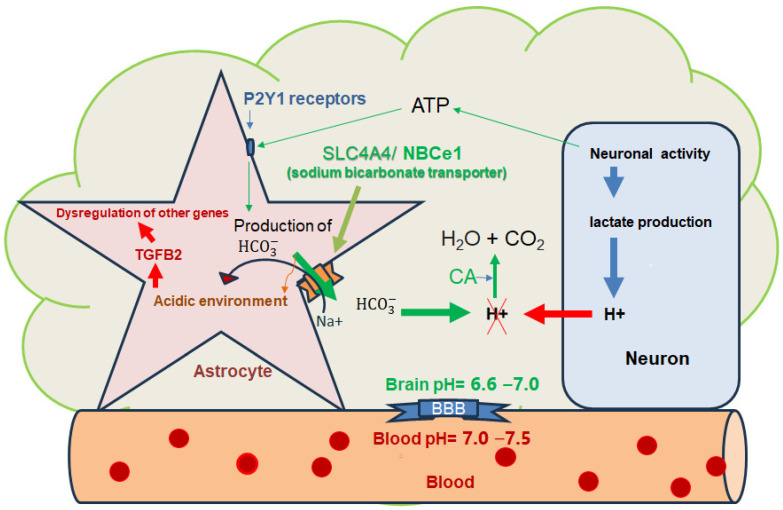
Neuronal activity is associated with lactate generation reducing brain pH. Astrocytes produce bicarbonate in response to neuronal activity-dependent ATP release stimulating P2Y1 receptors. SLC4A4 (sodium bicarbonate cotransporter 1, or Solute Carrier Family 4 Member 4) transports bicarbonate into the intercellular space (in exchange with Na^+^) to neutralize H^+^ leading to the formation of CO_2_ and water, a process mediated by carbonic anhydrase (CA). In return, the outward flow of bicarbonate reduces astrocytes’ pH compared to other brain cells. Note that the pH of the brain compartment is less than blood pH. While a study reported that lactate and exercise-induced lactate can increase TGFB2 (transforming growth factor beta-2) expression in adipose tissue, the decrease in pH mediated by lactate could be linked to an increase in TGFB2 expression in iPSC-derived astrocytes in vitro as well as in post-mortem brains of patients with SCZ (schizophrenia) and bipolar disorder (iPSC: induced pluripotent stem cells).

**Figure 2 biomedicines-12-00457-f002:**
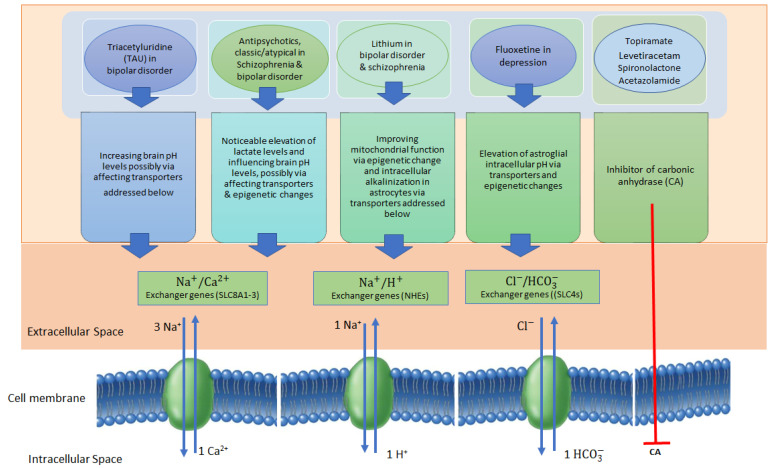
An overview of the mechanisms of action of antipsychotics and antidepressants via regulation of brain pH. There are several mechanisms by which antipsychotics and antidepressants may exert their beneficial effects on major mental disorders through the regulation of brain pH or brain lactate regulation. Some drugs confer neuroprotective effects by elevating pH levels or inducing intracellular alkalinization in astrocytes through the modulation of different exchangers (transporters), including the Na^+^/H^+^ exchangers (NHEs: SLC9A1-A3, SLC9A6-9), Na^+^/Ca^2+^ exchangers (SLC8A1-3), and Cl^−^/HCO_3_^−^ exchangers (e.g., SLC4A1-A3, SLC4A7 and 8, SLC4A10, SLC26A6). Other drugs, such as spironolactone, can act as inhibitors of carbonic anhydrase (CA), which catalyzes the conversion of carbon dioxide to bicarbonate and vice versa.

**Table 1 biomedicines-12-00457-t001:** Different mental diseases that are associated with reduced brain pH.

Diseases	Type of Study	pH Changes	Key Finding	Ref
Schizophrenia (SCZ)/bipolar disorder	Post-mortem brains	Lower pH levels (*p* < 0.05)	Reduced brain pH in patients with high inflammatory/stress vs. controls (attributed to tissue injury or response to an elevated metabolic demand)	[45]
Bipolar disorder	Clinical study	Lower intracellular pH (*p* < 0.05)	Association between altered cellular metabolism and reduced intracellular pH in the brains of patients	[46]
Bipolar disorder	Clinical study	Lower pH levels (*p* < 0.05)	Increasing 4-hydroxynonenal (4-HNE) due to lipid peroxidation in the anterior cingulate cortex, reducing pH in patients vs. controls	[47]
Bipolar disorder	Clinical study	Lower pH levels (*p* < 0.05)	Reduced pH in the anterior cingulate of unmedicated manic adolescents vs. healthy subjects	[48]
Bipolar disorder	Clinical study	Lower pH levels (*p* < 0.05)	Decreased pH levels, and mitochondrial dysfunction in patients	[49]
Depression and major depressive disorder (MDD)	Clinical study in MDD and experimental study in mice	Decreased carbonic anhydrase level (*p* < 0.05)	Decreased carbonic anhydrase 1 (CAR1) level in MDD; depression-like behaviors in CAR1-knockout mice due to lower extracellular bicarbonate (i.e., lower pH)	[50]
Manic-depressive or recurrent depression	Clinical study of saliva	Higher pH levels (*p* < 0·001)	Reduced membrane transport, stronger sodium activity, and higher pH levels vs. controls due to imbalances in sodium and bicarbonate reabsorption	[51]
Autism spectrum disorder (ASD)	Clinical study (saliva)	Lower resting pH levels (*p* < 0.05)	Lower resting pH of saliva in autistic children vs. healthy children	[52]
Alzheimer’s Disease (AD)	Clinical study (magnetic resonance spectroscopy)	Higher pH levels (*p* < 0.05)	Elevated pH to alkaline range in the left hippocampus of AD patients	[53]
AD	Experimental study (Human Aβ1–42)	Impact of pH on Aβ1–42 aggregation	Lack of Aβ_1–42_ aggregation at pH 9.5	[54]
AD	Clinical study	Lower pH levels (*p* < 0.05)	Lower pH in the hippocampus during normal aging and lower pH in periventricular white matter in AD	[55]
AD	Experimental study (mice)	Lower pH levels (*p* < 0.004)	Reducing brain and CSF pH in AD mice and increasing Aβ plaques load in APP-PS1 mice after CSF infusion with low pH	[56]
AD	Clinical study	Higher pH levels (*p* < 0.05)	Glutathione depletion in the left and right hippocampus and higher pH levels in the left hippocampus in AD vs. controls	[57]
AD	Research hypothesis	Higher pH levels	Association between the hippocampal GSH depletion and increased hippocampal pH levels in AD	[58]

**Table 2 biomedicines-12-00457-t002:** Studies that reported elevated lactate levels in different neuropsychiatric diseases.

Disease	Type of Study	Key Finding	Ref
Schizophrenia (SCZ)	Clinical study	Association between elevated lactate level in post-mortem brains of SCZ patients and reduced brain pH	[59]
SCZ	Clinical study	Higher levels of lactate and pyruvate, and lower levels of β subunit of pyruvate dehydrogenase, in the striatum of SCZ patients vs. controls	[60]
SCZ	Clinical study	Higher lactate level in patients vs. controls; cognitive deficits in patients because of elevated anaerobic glycolysis (likely due to mitochondrial dysfunction)	[35]
SCZ/bipolar disorder	Experimental study in mice; clinical study in the patients	Higher lactate level and hence lower pH in the brains of model mice vs. controls; lower brain pH in SCZ and bipolar disorder vs. control	[36]
SCZ	Clinical study in SCZ and experimental study in mice	Elevated lactate level in the dorsolateral prefrontal cortex in SCZ, and in iPSC-derived frontal cortical neurons of a SCZ patient with DISC1 mutation; reduced lactate level in astrocytes of mice with “induced expression of mutant human DISC1”	[61]
SCZ	Clinical study	Elevated blood level of lactate during exercise, and lower mitochondrial DNA copy numbers vs. control, indicating mitochondrial dysfunction in SCZ	[62]
SCZ	Experimental study in male adult rats	Association between high-level lactate production by astrocytes and deficits in auditory sensory gating in isolated rats	[63]
Bipolar disorder, depressed	Clinical study	Higher lactate level in the cingulate cortex of patients, which was decreased after 6-weeks lithium monotherapy	[64]
MDD	Clinical study	Elevated ventricular lactate in adolescents with MDD vs. controls	[65]
MDD	Clinical study	Decreased mitochondrial oxidative clearance of lactate, elevated glucose and lactate levels in patients, linked to increased depression severity	[66]
ASD	Clinical study	Higher lactate level and lactate-to-pyruvate ratio in ASD	[67]
ASD	Clinical study	Increased cerebral lactate level as a sign of mitochondrial dysfunction in adults with ASD	[68]
AD	Clinical study	Elevated level of the CSF lactate vs. controls	[69]
AD	Clinical study	Higher CSF lactate levels in earlier stages of AD	[70]

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
