# Peer review of "Potential for New Therapeutic Approaches by Targeting Lactate and pH Mediated Epigenetic Dysregulation in Major Mental Diseases"

_biomedicines, 2024, doi:10.3390/biomedicines12020457_

Round 1
Reviewer 1 Report
Comments and Suggestions for Authors
Review on the manuscript titled “Brain pH-Lactate and Epigenetic Dysregulation in Major Mental Diseases: Potentials for New Therapeutic Approach” by Nohesara et al., 2024
In their review manuscript the authors address the pH homeostasis alteration and coupled epigenetic regulations in mental diseases, as well as outlining possible therapeutic means for the phenomenon.
The authors state that H+/Na+ homeostasis dynamics alters energy metabolism via mitochondria dysfunction impact. The authors trace the ion metabolism pathways including Slc9a1, Slc4a4 ion transporters observed in a range of publications. Authors note that lactate level closely associate with pH level, and altering histone lactylation rate affecting histone methylation/acetylation, this way altering chromatin state in various activation sites. Brain disease studies describing the pH level impact on various processes are listed in Table 1, Table 2 features brain pathology studies with shifted lactate level and corresponding alteration of pH rate in brain disease clinical/animal studies.
In chapter 4, the authors underline the epigenetic modulation by lactylation of several histone marks that become a subject for further acetliation, which modify transcriptional properties of a range of genes reported in publications connected with psychiatric diseases, stressing the importance of lactate expression rate variation along with pH.
Chapter 5 describes pH/ion alterations of antipsychotic drugs via epigenetic changes such as histone H3K18 lactylation/acetylation. The authors draw an overview of the mechanisms of action of antipsychotics/antidepressants via regulation of brain pH. (Fig. 4), which looks like the major take out message of the manuscript.
In chapter 6 the authors introduce the gut microbiota share of lactate turnover, with some details and evidence from the publications.
The chapter 7 appeals to utilizing pH modulation strategy via the issues listed in the manuscript.
Overall, the review would be of interest to the professionals and therapists in the field. The authors thoroughly retrieved the corresponding publications (146 refs), and posited the vital importance of pH modulation in brain diseases as well as possible remedies existed today.
There are some notes on the manuscript below.
1. pH dynamics is rather of homeostatic nature with certain feedback loops. Its manual/therapeutic (long term) targeting may provide adverse effects as well.
2. While assessing post mortem samples pH, is there any assurance it is not shifted greatly in the first hours after death? How stable is pH variation?
3. Would microbiome lactate metabolism affect brain pH by lactate absorbance? What about BBB?
4. Table 2 “Experimental study in male adult male rats” – lot of ‘male’
5. P8:222-223:” MeCP2 is a key protein with binding capacity to various types of methylated DNA in the brain which its mutations highly influences important cellular” - spelling problem.
Comments on the Quality of English Language
There are few semantical errors (repeated words, hangup phrases) that need a cleanup across the manuscript.
Author Response
Reviewer report
In their review manuscript the authors address the pH homeostasis alteration and coupled epigenetic regulations in mental diseases, as well as outlining possible therapeutic means for the phenomenon.
The authors state that H+/Na+ homeostasis dynamics alters energy metabolism via mitochondria dysfunction impact. The authors trace the ion metabolism pathways including Slc9a1, Slc4a4 ion transporters observed in a range of publications. Authors note that lactate level closely associate with pH level, and altering histone lactylation rate affecting histone methylation/acetylation, this way altering chromatin state in various activation sites. Brain disease studies describing the pH level impact on various processes are listed in Table 1, Table 2 features brain pathology studies with shifted lactate level and corresponding alteration of pH rate in brain disease clinical/animal studies.
In chapter 4, the authors underline the epigenetic modulation by lactylation of several histone marks that become a subject for further acetylation, which modify transcriptional properties of a range of genes reported in publications connected with psychiatric diseases, stressing the importance of lactate expression rate variation along with pH.
Chapter 5 describes pH/ion alterations of antipsychotic drugs via epigenetic changes such as histone H3K18 lactylation/acetylation. The authors draw an overview of the mechanisms of action of antipsychotics/antidepressants via regulation of brain pH. (Fig. 4), which looks like the major take out message of the manuscript.
In chapter 6 the authors introduce the gut microbiota share of lactate turnover, with some details and evidence from the publications.
The chapter 7 appeals to utilizing pH modulation strategy via the issues listed in the manuscript.
Overall, the review would be of interest to the professionals and therapists in the field. The authors thoroughly retrieved the corresponding publications (146 refs), and posited the vital importance of pH modulation in brain diseases as well as possible remedies existed today.
Author response and action taken: We are grateful to the reviewer for the thorough evaluation of the manuscript and for providing constructive comments to enhance the quality of this work. We have carefully revised the manuscript, and the corresponding modifications in response to the reviewer's feedback are highlighted in yellow.
There are some notes on the manuscript below.
- pH dynamics is rather of homeostatic nature with certain feedback loops. Its manual/therapeutic (long term) targeting may provide adverse effects as well.
Author response and action taken: Thanks for the productive comment. We added your comment on page 14, lines 399-401.
- While assessing post mortem samples pH, is there any assurance it is not shifted greatly in the first hours after death? How stable is pH variation?
Author response and action taken: A study analyzing postmortem intervals and storage period concluded that pH is stable after death (0–48 h) and during freezer storage.
We included the result of this study on page 5, lines 163-166.
It is also noteworthy that based on our data, postmortem and refrigeration intervals may cause an increase (but not decrease) brain PH. In fact, in our sample set, there was a direct correlation between refrigeration intervals and brain pH in SCZ (r= 0.38, p=0.024, two tail t test). Similarly, there was a direct correlation between PMI and pH (r=0.36, p=0.037, two tailed t test). Therefore, the impact of low brain pH on gene expression alterations in SCZ may become subject to underestimation in the patients.
- Would microbiome lactate metabolism affect brain pH by lactate absorbance? What about BBB?
Author response and action taken: A study showed that lactate can be absorbed into the blood stream; cross the BBB and change brain pH. This is addressed on page 13, line 363-367.
- Table 2 “Experimental study in male adult male rats” – lot of ‘male’
Author response and action taken: Usually such studies are done in male rats or mice, because female animals may have variations due to their menstrual cycles. Therefore, almost all studies addressed in this work are in male animals.
- P8:222-223:” MeCP2 is a key protein with binding capacity to various types of methylated DNA in the brain which its mutationshighly influences important cellular” - spelling problem.
Author response and action taken: Thanks, we corrected errors (page 9, lines 210-211).
Comments on the Quality of English Language
There are few semantical errors (repeated words, hangup phrases) that need a cleanup across the manuscript.
Author response and action taken: Thanks for your constructive feedback. Based on your comments, the entire manuscript has been meticulously revised for grammar, punctuation, and spelling mistakes. Additionally, several sentences have been rephrased for better clarity. All such changes are written in green.
Reviewer 2 Report
Comments and Suggestions for Authors
GENERAL COMMENTS
This review covers an interesting topic outside the usual trodden paths. The schematic diagrams and table 1 are instructive and concise.
Table 1 would profit from stating the actual pH values as well as their measures of variance and declaring if the difference was statistically significant or not.
The text is very long.
Overall, the authors should please make up their mind whether consistently use upper or lower case, e.g., in the upper rightmost corner of figure 2: Topiramate (i.e., upper case) vs levetiracetam (lower case).
SPECIFIC ITEMS
line 19: The authors should please replace “article” with “review”.
line 19: Please delete “about”
Figure 1 legend: What do the authors mean by “While in adipse tissue …” Please rephrase and clarify. Also, for the reader who hates text and loves diagrams, please do not give only the abbreviations but spell out the terms as well.
Figure 2: If spread across a 27-inch screen, the letters become fuzzy. Please use a higher resolution for the figure. Please use lower case for “Astrocytes” and “Astroglial”.
Comments on the Quality of English Language
moderate editing advisable
Author Response
Reviwer report:
This review covers an interesting topic outside the usual trodden paths. The schematic diagrams and table 1 are instructive and concise.
The authors highly appreciate your kind consideration of this work. We have revised the manuscript based on your valuable comments, making the appropriate modifications and adding more information, all of which are highlighted in green.
Table 1 would profit from stating the actual pH values as well as their measures of variance and declaring if the difference was statistically significant or not.
Author response and action taken: All studies summarized in table 1 addressed statistically significant alterations. However, actual pH values and their measures of variance are not mentioned in all studies. Therefore, to present the data uniformly, we added all p values to table 1. Additionally, whenever indicated, the actual pH values and standard deviations (SD) are presented in multiple locations in the text
-The text is very long.
Author response and action taken: We revised Table 1 and Table 2 and summarized their text (written in green color).
Overall, the authors should please make up their mind whether consistently use upper or lower case, e.g., in the upper rightmost corner of figure 2: Topiramate (i.e., upper case) vs levetiracetam (lower case).
Author response and action taken: We revised figure 2 in response to this important comment.
SPECIFIC ITEMS
-line 19: The authors should please replace “article” with “review”.
Author response and action taken: It has been corrected. Thanks.
Page and line number(s) for changes made: (page 1, line 19).
- line 19: Please delete “about”
Author response and action taken: Thanks, it has been deleted.
Page and line number(s) for changes made: (page 1, line 19).
Figure 1 legend: What do the authors mean by “While in adipose tissue …” Please rephrase and clarify. Also, for the reader who hates text and loves diagrams, please do not give only the abbreviations but spell out the terms as well.
Author response and action taken: When considering adipose tissue, the original study (Takahashi et al., 2019) focused on examining the effects of lactate and exercise induced lactate on adipose tissue. However, our preliminary data, as discussed in the paragraph below this figure (lines 143-157), indicated that lactate effect on TGFB2 expression is not limited to adipose tissue. Therefore, we have rephrased the legend of Fig. 1 to clarify this issue.
Abbreviations are also spelled out in tables and figures.
Page and line number(s) for changes made: Figure 1 legend.
-Figure 2: If spread across a 27-inch screen, the letters become fuzzy. Please use a higher resolution for the figure. Please use lower case for “Astrocytes” and “Astroglial”.
Author response and action taken: We used a higher resolution for the figure and lower case for “Astrocytes” and “Astroglial”.
Page and line number(s) for changes made: Figure 2
Thanks again for all constructive comments.
Reviewer 3 Report
Comments and Suggestions for Authors
Over all this is manuscript will add a new knowledge needed in the filed.
It is recommended to change the title as follows:
Potentials New Therapeutic Approaches for Brain pH-Lactate and Epigenetic Dysregulation in Major Mental Diseases
It is advised to improve the scientific content quality of figure 1.
Please add DOI links for all cited references.
There are redundancy in the cited references and in order to limit the large number of the cited references, please omit the old references for example from 1997 to 2005.
Author Response
Reviewer report:
Comments and Suggestions for Authors
-Over all this is manuscript will add a new knowledge needed in the field.
The authors are deeply grateful for your appreciation of this work. We have revised the manuscript based on your important comments, making the appropriate modifications and adding more information, all of which are highlighted in blue.
-It is recommended to change the title as follows: Potentials New Therapeutic Approaches for Brain pH-Lactate and Epigenetic Dysregulation in Major Mental Diseases
Author response and action taken: we changed the title based on your comment with minor modification as below:
“Potentials New Therapeutic Approaches for pH-Lactate-Induced Brain Epigenetic Dysregulation in Major Mental Diseases”
However, we are open to accepting your proposed title if the modified title is inappropriate.
Page and line number(s) for changes made: page 1, the title
-It is advised to improve the scientific content quality of figure 1.
Author response and action taken: We improved the scientific quality of figure 1 and added new elements to this figure.
Page and line number(s) for changes made: figure 1.
-Please add DOI links for all cited references.
Author response and action taken: We added DOI links for all cited references, whenever available.
Page and line number(s) for changes made: references
-There are redundancy in the cited references and in order to limit the large number of the cited references, please omit the old references for example from 1997 to 2005.
Author response and action taken: We removed approximately 10 redundant citations, while retaining single citations from the years 1997 to 2005 and those addressed in Tables 1 and 2, as we aimed to provide a comprehensive review. It's important to note that in response to other reviews, we added new references, so the total number of cited references has not changed.